# An Interpretable Representation Learning Approach for Diffusion Tensor Imaging

**Vishwa Mohan Singh**[*1,2] (ID)                    Vishwa.Singh@med.uni-muenchen.de
**Alberto Gaston Villagran Asiares**[2]              alberto.villagran@med.uni-muenchen.de
**Luisa Sophie Schuhmacher**[2]                      luisa.schuhmacher@med.uni-muenchen.de
**Kate Rendall**[*2]                                 kate.rendall@med.uni-muenchen.de
**Simon Weißbrod**[*2]                               simon.weissbrod@med.uni-muenchen.de
**David Rügamer**[1,3]                               david.ruegamer@stat.uni-muenchen.de
**Inga Körte**[2,4,5]                                inga.koerte@med.uni-muenchen.de

[1] *Department of Statistics, Ludwig-Maximilians-Universität München, Germany*

[2] *cBRAIN, Department of Child and Adolescent Psychiatry, Psychosomatics, and Psychotherapy, Ludwig-Maximilians-Universität München, Germany.*

[3] *Munich Center for Machine Learning, Munich, Germany.*

[4] *Psychiatry Neuroimaging Laboratory, Mass General Brigham Academic Medical Centers, Psychiatry Department, Boston, MA, USA.*

[5] *Harvard Medical School, Boston, MA, USA*

**Editors:** Accepted for publication at MIDL 2025

## Abstract

Diffusion Tensor Imaging (DTI) tractography offers detailed insights into the structural connectivity of the brain, but presents challenges in effective representation and interpretation in deep learning models. In this work, we propose a novel 2D representation of DTI tractography that encodes tract-level fractional anisotropy (FA) values into a 9×9 grayscale image. This representation is processed through a Beta-Total Correlation Variational Autoencoder ($\beta$-TCVAE) with a Spatial Broadcast Decoder to learn a disentangled and interpretable latent embedding. We evaluate the quality of this embedding using supervised and unsupervised representation learning strategies, including auxiliary classification, triplet loss, and SimCLR-based contrastive learning. Compared to the 1D Group deep neural network (DNN) baselines, our approach improves the F1 score in a downstream sex classification task by 12.64% and shows a better disentanglement than the 3D representation.

**Keywords:** Diffusion Tensor Imaging, Autoencoders, Representation Learning

## 1. Introduction

Diffusion Tensor Imaging (DTI) tractography is a non-invasive technique that models white matter fiber bundles in the brain (Buyanova and Arsalidou, 2021; Jelescu and Budde, 2017). It has become increasingly crucial for studying neurodevelopment, aging, and neurological diseases (Sundgren et al., 2004). However, effectively representing the complex geometry and connectivity information in DTI tractography remains a challenge in analysis. Methodologies using 1-dimensional representation often disregard spatial context, while the complex 3D architecture lacks interpretability (Related works in Appendix B).

---

* Contributed equally

To address this, we propose a novel 2D representation of DTI tractography that maintains critical spatial information while remaining amenable to deep learning techniques. Specifically, we transform tract-level fractional anisotropy (FA) values into a $9\times9$ grid format, where each pixel encodes the FA value of one tract. From this grid, we learn a disentangled class-aware representation using a combination of a Disentangled VAE and representation learning strategies. This representation is meant to be used in a late-fusion multi-modal architecture to analyze different MRI Modalities.

## 2. Methodology

### 2.1. Dataset and Representation

Data was collected from young adult amateur soccer players with at least 5 years of organized training (46 males and 23 females), and control athletes engaged in non-contact sports (11 males and 25 females). On this, we use sex classification as our downstream task. For postprocessing, we use WMA800 (O'Donnell and Westin, 2007; O'Donnell et al., 2012) to divide the white matter fibers into 74 different anatomical tracts based on the ORG-800FC-100HCP atlas (Zhang et al., 2018).

We convert this WMA output to a compact 2D representation of DTI tractography. The 74 tracts are arranged in a $9\times9$ grid using Multi-Dimensional Scaling (Mead, 1992) to preserve distance. After finding the grid coordinates, we use the Hungarian algorithm to solve the overlap between several centroids (See Appendix Algorithms 1 and 2).

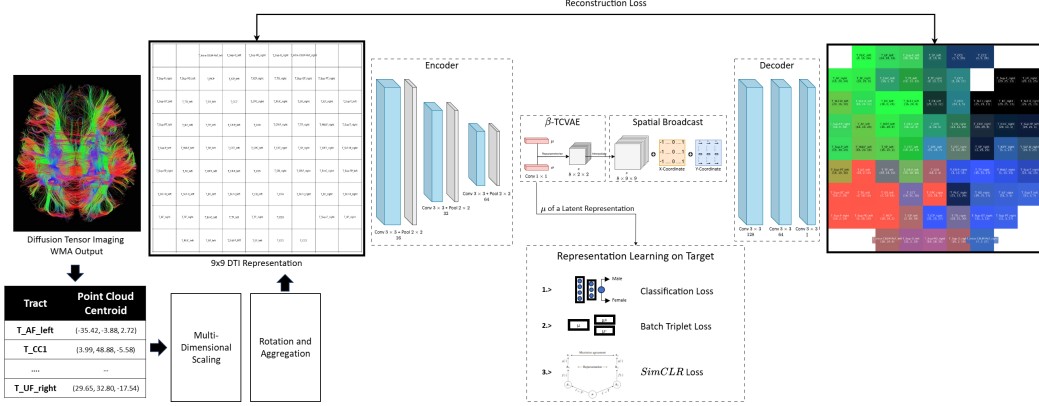

Figure 1: Architecture to convert the Dense DTI representation to an interpretable latent vector $Z$. The representation learning and classification are performed on the estimated mean $mu$. The reconstruction shows latent values, which are the explainers for each tract.

### 2.2. Autoencoder and Representation Learning

The core of our model is a variational autoencoder (Kingma et al., 2013). The encoder compresses the $9\times9$ FA image into a latent vector of size 32 and a spatial broadcast de-

coder (Watters et al., 2019) to make the latent vector retain spatial coherence. The model is trained using a $\beta$-TCVAE loss, which encourages disentanglement by penalizing total correlation (Chen et al., 2018). To retain class-relevant information in the latent space, we evaluate three strategies: a supervised auxiliary classifier as a proxy for semantic structure, a triplet loss for preserving local class-wise distances (Hoffer and Ailon, 2015), and an unsupervised SimCLR loss for learning global latent structure (Chen et al., 2020). The full architecture is shown in Figure 1.

## 3. Experiment and Results

The following Table 1 shows the results from the models tested. The Disentangled (Dis.) VAE in the table refers to the model using both spatial broadcasting and $\beta$-TCVAE loss. As controls, we use a 1D deep neural network (DNN), a Grouped DNN, and a 3D Autoencoder, which works on the original centroid position. We compare the separability (Sep) of the latent space using a KNN classifier (Dyballa et al., 2024) with $k = 3$ and the metrics from the best classifier from LazyClassifier (Pandala and Silva, 2019). We measure the Mutual Information Gap (MIG) (Chen et al., 2018) of different regions to evaluate disentanglement.

Table 1: Comparisons of Models over classification performance, reconstruction, and Mutual Information Gap

| Network | Accuracy | F1 | Sep | Recon | MIG |
|---|---|---|---|---|---|
| 1D-DNN | 53.90 ($\pm$ 12.2) | 51.15 ($\pm$ 28.4) | - | - | - |
| 1D Group DNN | 65.00 ($\pm$ 14.0 ) | 68.78 ($\pm$ 15.7) | - | - | - |
| 3D VAE + Aux | **82.60 ($\pm$ 9.4)** | **82.06 ($\pm$ 9.7)** | 77.08 | 0.0190 | 0.0344 |
| 2D VAE + Aux | 80.90 ($\pm$ 9.5) | 80.15 ($\pm$ 10.1) | 81.25 | 0.0159 | 0.0503 |
| 2D $\beta$-TCVAE + Aux | 81.45 ($\pm$ 13.5) | 81.37 ($\pm$ 13.4) | **83.33** | **0.0151** | 0.0535 |
| 2D Dis. VAE + Aux | 80.09 ($\pm$ 8.6) | 79.74 ($\pm$ 9.5) | 81.25 | 0.0180 | 0.0640 |
| 2D Dis. VAE + Triplet | 77.27 ($\pm$ 8.6) | 77.30 ($\pm$ 8.1) | 81.25 | 0.0171 | **0.0739** |
| 2D Dis. VAE + SimCLR | 81.72 ($\pm$ 12.0) | 81.42 ($\pm$ 12.3) | 70.81 | 0.0768 | 0.0588 |

## 4. Discussion and Conclusion

Along with better disentanglement than 3D, the 2D representation improves over the best 1D model by 12.64% in F1 score, with no significant drop from the 3D equivalent. Interpretation results from SHAP (Lundberg and Lee, 2017) show that across the subjects, the male subjects show a higher FA value, especially in the left Corona Radiata, Right Superior Longitudinal Fasciculus, and Right Corticospinal Tracts (refer Appendix E Figure 2). Some of these align with the findings from previous DL and statistical analyses (Menzler et al., 2011; Chen et al., 2023).

Further improvements can be made to the architecture's interpretability by finding a better balance of classification and the Kullback-Leibler term. Additionally, methods like attention (Vaswani et al., 2017) or factorization machines (Rendle, 2010) could be used to model any interactions between the tracts.

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

## Appendix A. Availability of Code

The code used to generate these results is in the following anonymized public repository:
https://github.com/SAint7579/DTI_2D_representation

## Appendix B. Related Literature

The most common way to pair tractography output with machine learning has been 1D feature-based approaches using models like support vector machines (Irimia et al., 2018). These methods are computationally efficient and interpretable, but suffer from the loss of spatial and geometric information. TractGraphCNN (Chen et al., 2023) tries to address this by rearranging the fiber bundles in a graph. More expressive 3D alternatives like TRAFIC (Lam et al., 2018) and Deep White Matter Analysis (Zhang et al., 2020) representations, on the other hand, make the model harder to interpret.

Autoencoders, especially Variational Autoencoders (VAEs) (Kingma et al., 2013), have recently been used to learn low-dimensional embeddings from tractography data (Feng et al., 2023; Trinkle et al., 2021). Furthermore, studies integrating DTI with other modalities (e.g., fMRI, EEG) have demonstrated the importance of compact and interpretable embeddings for multimodal fusion (Qu et al., 2024; Zhang et al., 2021).

## Appendix C. Mathematical Definitions of the Loss Function

The training objectives in our framework are built on the $\beta$-Total Correlation Variational Autoencoder ($\beta$-TCVAE) backbone, with three distinct variants depending on the auxiliary objective. The base $\beta$-TCVAE loss consists of a reconstruction term and a decomposed KL divergence that includes mutual information (MI), total correlation (TC), and dimension-wise KL. The total loss is given by:

$$\mathcal{L}_{\text{TCVAE}} = \mathcal{L}_{\text{recon}} + \mathcal{L}_{\text{KL}}, \tag{1}$$

where the reconstruction loss is defined as the mean squared error between the input $x$ and the reconstruction $\hat{x}$:

$$\mathcal{L}_{\text{recon}} = \|x - \hat{x}\|_2^2. \tag{2}$$

The KL divergence is decomposed as:

$$\mathcal{L}_{\text{KL}} = \underbrace{\text{MI}(z; x)}_{\text{mutual information}} + \beta \cdot \underbrace{\text{TC}(z)}_{\text{total correlation}} + \underbrace{\sum_j D_{\text{KL}}(q(z_j)\|p(z_j))}_{\text{dimension-wise KL}}, \tag{3}$$

where $\beta$ controls the strength of the disentanglement by scaling the total correlation term.

**1. Auxiliary Classifier Loss:** To guide the latent space with supervised signals, we add a binary cross-entropy loss from an auxiliary classifier $f_{\text{cls}}$ operating on the latent mean $\mu$. The total loss becomes:

$$\mathcal{L}_{\text{AE+Cls}} = \lambda_{\text{VAE}} \cdot \mathcal{L}_{\text{TCVAE}} + \lambda_{\text{Cls}} \cdot \mathcal{L}_{\text{BCE}}(f_{\text{cls}}(\mu), y), \tag{4}$$

where $y$ is the ground truth label and $\lambda_{\text{Cls}}$ balances the classification loss.

**2. Triplet Loss:** To structure the latent space based on local class similarity, we apply a batch-hard triplet loss on the latent mean $\mu$, enforcing separation between positive and negative pairs:

$$\mathcal{L}_{\text{AE+Triplet}} = \lambda_{\text{VAE}} \cdot \mathcal{L}_{\text{TCVAE}} + \lambda_{\text{Triplet}} \cdot \mathcal{L}_{\text{Triplet}}(\mu, y), \tag{5}$$

where $\mathcal{L}_{\text{Triplet}}$ is defined as:

$$\mathcal{L}_{\text{Triplet}} = \max\left(0, \|\mu_a - \mu_p\|_2^2 - \|\mu_a - \mu_n\|_2^2 + \alpha\right), \tag{6}$$

with anchor $\mu_a$, positive $\mu_p$, negative $\mu_n$, and margin $\alpha$.

**3. SimCLR Contrastive Loss:** For unsupervised structure in the latent space, we apply the SimCLR loss on pairs of augmentations $x_1$, $x_2$ passed through the encoder, using the latent mean $\mu$ as the representation. The combined loss is:

$$\mathcal{L}_{\text{AE+SimCLR}} = \lambda_{\text{VAE}} \cdot \mathcal{L}_{\text{TCVAE}} + \lambda_{\text{SimCLR}} \cdot \mathcal{L}_{\text{SimCLR}}(\mu_1, \mu_2), \tag{7}$$

where the SimCLR loss is defined as:

$$\mathcal{L}_{\text{SimCLR}} = -\log \frac{\exp(\text{sim}(\mu_1, \mu_2)/\tau)}{\sum_{j=1}^{2N} 1_{[j \neq i]} \exp(\text{sim}(\mu_i, \mu_j)/\tau)}, \tag{8}$$

with cosine similarity $\text{sim}(\cdot, \cdot)$ and temperature parameter $\tau$.

Each of these loss formulations guides the latent representation toward a specific structure—semantic separability, local neighborhood coherence, or augmentation invariance—while maintaining reconstruction quality and disentanglement through the $\beta$-TCVAE framework.

## Appendix D. Algorithm for Rearrangement

To create a compact and spatially meaningful 2D representation of DTI tractography, we project 3D tract centroids onto a 2D grid. Algorithm 1 outlines the procedure, which first applies Multi-Dimensional Scaling (MDS) to preserve inter-tract distances, followed by normalization to fit the projected coordinates within a 9×9 grid. This forms the basis for consistent tract placement across subjects.

---

**Algorithm 1:** Convert DTI 3D Representation to 2D Grid

---

**Input:** $X = \{x_1, x_2, \ldots, x_n\}$, where $x_i \in \mathbb{R}^3$ (3D data points)

**Output:** $G = \{g_1, g_2, \ldots, g_n\}$, where $g_i \in \{1, \ldots, 9\} \times \{1, \ldots, 9\}$ (2D grid positions)

**Step 1: Dimensionality Reduction via MDS**;

$Y \leftarrow \text{MDS}(X, d = 2)$;

**Step 2: Normalize 2D Coordinates to a 9x9 Grid**;

Compute $y_{\min,1}$ and $y_{\max,1}$, the minimum and maximum of the first coordinates in $Y$;

Compute $y_{\min,2}$ and $y_{\max,2}$, the minimum and maximum of the second coordinates in $Y$;

**for** *each point* $y_i = (y_{i,1}, y_{i,2}) \in Y$ **do**

$\quad \left|\quad g_{i,1} \leftarrow \text{round}\left(\dfrac{y_{i,1} - y_{\min,1}}{y_{\max,1} - y_{\min,1}} \times 8\right) + 1;\right.$

$\quad \left|\quad g_{i,2} \leftarrow \text{round}\left(\dfrac{y_{i,2} - y_{\min,2}}{y_{\max,2} - y_{\min,2}} \times 8\right) + 1;\right.$

$\quad \left|\quad g_i \leftarrow (g_{i,1}, g_{i,2});\right.$

**end**

**Step 3: Rearrangement using the Hungarian Algorithm**;

Construct a cost matrix $C \in \mathbb{R}^{n \times n}$ where $C(i, j)$ is the distance between $g_i$ and the $j$th grid position;

$P \leftarrow \text{HungarianAlgorithm}(C)$;

Reassign each point $y_i$ to the grid position indicated by $P$;

**return** $G$;

---

To resolve overlapping grid positions resulting from the MDS projection, we use the Hungarian Algorithm to optimally assign tracts to unique 2D locations while minimizing displacement. Algorithm 2 summarizes this process, ensuring a one-to-one mapping of tracts to grid positions with minimal distortion of spatial relationships.

---

**Algorithm 2:** HungarianAlgorithm

---

**Input:** $C \in \mathbb{R}^{n \times n}$, the cost matrix
**Output:** $P$, the optimal assignment mapping each row to a column

**Step 1: Row Reduction**;
**for** $i \leftarrow 1$ **to** $n$ **do**
  $minRow \leftarrow \min\{C(i,j) : 1 \leq j \leq n\}$;
  **for** $j \leftarrow 1$ **to** $n$ **do**
  | $C(i,j) \leftarrow C(i,j) - minRow$;
  **end**
**end**

**Step 2: Column Reduction**;
**for** $j \leftarrow 1$ **to** $n$ **do**
  $minCol \leftarrow \min\{C(i,j) : 1 \leq i \leq n\}$;
  **for** $i \leftarrow 1$ **to** $n$ **do**
  | $C(i,j) \leftarrow C(i,j) - minCol$;
  **end**
**end**

**Step 3: Cover Zeros with Minimum Number of Lines**;
Cover all zeros in $C$ using the minimum number of horizontal and vertical lines;

**Step 4: Test for Optimality**;
**if** *the number of covering lines equals $n$* **then**
  | **return** the optimal assignment $P$ determined from the positions of the zeros in $C$;
**end**
**else**
  **Step 5: Adjust the Matrix**;
  Find the smallest uncovered value $k$ in $C$;
  **for** *each element $C(i,j)$ that is **not** covered by any line* **do**
  | $C(i,j) \leftarrow C(i,j) - k$;
  **end**
  **for** *each element $C(i,j)$ that is covered **twice** (i.e., by both a row and a column)* **do**
  | $C(i,j) \leftarrow C(i,j) + k$;
  **end**
  **Return to Step 3**;
**end**

---

## Appendix E. SHAP Results

The following Figure 2 shows the SHAP results for sex classification on tracts.

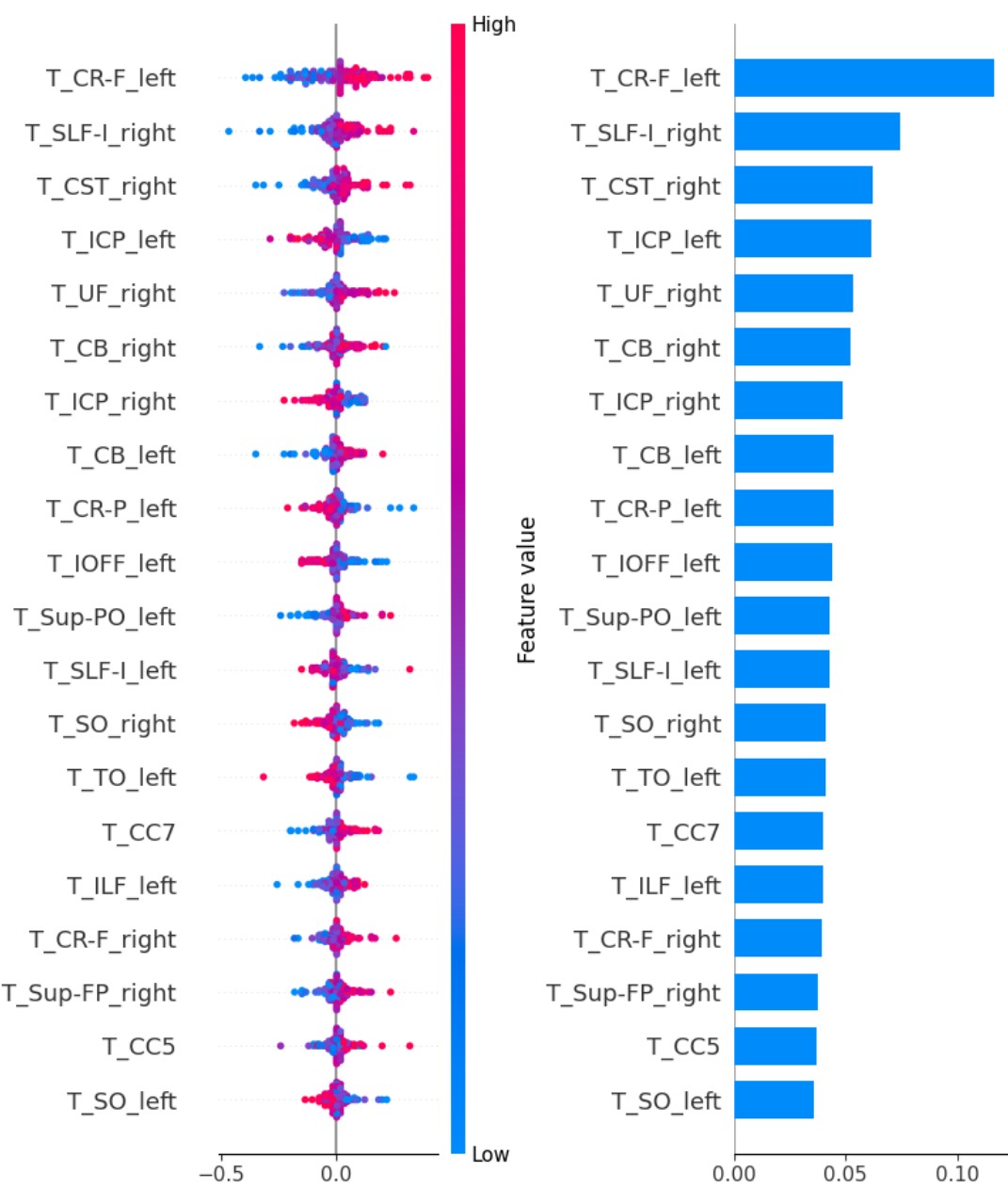

Figure 2: Graph that shows the impact of a tract (right) and the polarity of the interaction with the target i.e. sex classification (left).

