# OpenReview forum: "An Interpretable Representation Learning Approach for Diffusion Tensor Imaging"
_MIDL.io/2025/Short_Papers — MIDL 2025 - Short Papers_

### Official Review · Reviewer_yRrn · 2025-04-28

**Rating:** 4
**Confidence:** 5

**Summary:**

This paper presents a novel latent representation of DTI tractography to encode tract-level fractional anisotropy (FA) values. Specifically, the authors propose using Multi-Dimensional Scaling (MDS) to represent the FA map as a 9×9 grid, followed by the application of a Beta-Total Correlation Variational Autoencoder (β-TCVAE) to learn the latent representation.

**Strengths:**

The idea of developing an interpretable latent representation of the FA map is relatively novel.

**Weaknesses:**

The use of a 9×9 grid may oversimplify the original data, potentially leading to information loss. Consequently, the utility of learning a latent representation from an already highly compressed 9×9 grid is unclear. Furthermore, the method is only compared to simple baselines or minor variations of the proposed model, lacking comparisons with state-of-the-art methods.

---

### Decision · Program_Chairs · 2025-05-01

Accept